# GitTables: A Large-Scale Corpus of Relational Tables

**Madelon Hulsebos**[1,2], **Çağatay Demiralp**[1], and **Paul Groth**[2]

[1]Sigma Computing, San Francisco, CA 94105
[2]University of Amsterdam, Amsterdam, 1012 WX

{madelon,cagatay}@sigmacomputing.com
p.t.groth@uva.nl

## Abstract

The practical success of deep learning has sparked interest in improving relational table tasks, like data search, with models trained on large table corpora. Existing corpora primarily contain tables extracted from HTML pages, limiting the capability to represent offline database tables. To train and evaluate high-capacity models for applications beyond the Web, we need additional resources with tables that resemble relational database tables.

Here we introduce GitTables, a corpus of currently 1.7M relational tables extracted from GitHub. Our continuing curation aims at growing the corpus to at least 10M tables. We annotate table columns in GitTables with more than 2K different semantic types from Schema.org and DBpedia. Our column annotations consist of semantic types, hierarchical relations, range types and descriptions. The corpus is available at `https://gittables.github.io`.

Our analysis of GitTables shows that its structure, content, and topical coverage differ significantly from existing table corpora. We evaluate our annotation pipeline on hand-labeled tables from the T2Dv2 benchmark and find that our approach provides results on par with human annotations. We demonstrate a use case of GitTables by training a semantic type detection model on it and obtain high prediction accuracy. We also show that the same model trained on tables from the Web generalizes poorly.

## 1 Introduction

Deep learning (DL) models in the past decade have dramatically improved many long standing computer vision and natural language processing (NLP) tasks [22]. This practical success of DL has also spurred interest in its applications to tasks in other domains, including data management tasks, from data cleaning to annotation [39]. To train DL models, earlier work primarily relied on corpora consisting of tables scraped from HTML pages [43], such as WebTables [8] and WikiTables [4]. These corpora have the scale for training high capacity models and are rich in textual values, making them attractive for applications of existing DL models, particularly NLP models. As such, they have played an important role in facilitating research into data analysis and management tasks [7, 43].

However, tables extracted from HTML pages on the Web (Web tables) provide at best a skewed representation of tables in the wild residing in databases, particularly enterprise databases [24, 11, 21]. For example, the semantic type `id` or `identifier` does not even appear among the twenty most frequent headers of WebTables [40], the largest table corpus to date. It is therefore unsurprising that models based on Web tables have limited applications beyond the Web [7]. To broaden the impact of data-driven data management research, we need new tabular data collections complementing existing corpora with tables resembling database tables.

Preprint. Under review.

| id / Isolate Id | Study | species / Species | element group / Organism Group | country / Country | state / State | gender / Gender | age / Age Group | specialty / Speciality | source / Source | number of person born in place / In/Out Patient | year / Year |
|---|---|---|---|---|---|---|---|---|---|---|---|
| 1308726 | TEST | Enterococcus faecium | Enterococcus spp | Vietnam | nan | Male | 19 to 64 Years | Medicine General | GU: Urine | Inpatient | 2015 |
| 1308726 | TEST | Enterococcus faecium | Enterococcus spp | Vietnam | nan | Male | 19 to 64 Years | Medicine General | GU: Urine | Inpatient | 2015 |
| 1308726 | TEST | Enterococcus faecium | Enterococcus spp | Vietnam | nan | Male | 19 to 64 Years | Medicine General | GU: Urine | Inpatient | 2015 |

Figure 1: An example of an annotated table in GitTables. Annotations are provided with confidence scores. The match between "Species" and `species` has, for example, a score of 1, while "Organism Group" to `element group` has a score of 0.7. These scores enable users to filter annotations based on their use-case.

To address the demand for such a collection, we introduce GitTables: a corpus with 1.7M relational tables extracted from GitHub files in the comma-separated values (CSV) format. We will keep the corpus growing to have at least 10M tables. Large-scale data repositories have enabled pretrained language models such as BERT [14] and GPT [6]. The scale of GitTables will facilitate the extension of deep transfer learning models to downstream tasks such as data preparation, analysis and discovery.

Given the wide usability of column semantics for such tasks, we also annotate table columns using a syntactic as well as semantic method with the DBpedia and Schema.org ontologies (an example in Figure 1). Semantically rich annotations together with DL present a unique opportunity to model table semantics as demonstrated by TURL [13], TabFact [10], and TaBERT [41]. Therefore, our annotations include semantic types of columns along with their range type, descriptions, and hierarchical relations.

Our analysis of GitTables confirms the different nature of the tables: the tables in our corpus have significantly larger dimensions (rows and columns) and structurally different content than Web tables. The semantic type distribution also deviates significantly from the semantic distribution of Web tables, illustrating its different topical coverage. GitTables is available through `https://gittables.github.io`.

In summary, we contribute (1) GitTables, a new large-scale table corpus. To the best of our knowledge, GitTables is the first large-scale relational table corpus with topical coverage and content structurally different than tables extracted from HTML pages. (2) A new scalable automated column-annotation method using distant-supervision. We annotate the columns in GitTables with semantics consisting of semantic types, range types, hierarchical relations, and descriptions, making GitTables the largest annotated table corpus to date. (3) An analysis of GitTables, illustrating its complementary characteristics to existing corpora, as well as an experiment demonstrating its value for learning semantic column type models.

## 2 Related work

Web initiatives such as Common Crawl, Wikipedia, and Open Data have been cost-effective resources for curating unstructured and structured data at scale [27, 4, 28]. Below we discuss large-scale table corpora sourced from these initiatives and review prior work annotating column semantics of tables sampled from these corpora in.

### 2.1 Large-scale table corpora

**WebTables and Dresden Web Tables** WebTables and the Dresden Web Tables corpora extract tables from HTML pages in the Common Crawl corpus [8, 15]. These corpora provide an abundance of relational tables ranging from 59M to 90M and have been instrumental in advancing applications like table augmentation and integration [7, 43]. However, as interest in relational tables beyond the Web increases, it is clear that Web tables generalize poorly due to their small dimensions and different content [21].

**WikiTables** To provide high-quality tables with semantics that are easier to detect than those of arbitrary Web tables, WikiTables extracts approximately 2M tables from Wikipedia [4]. This corpus is primarily suitable for tasks such as question answering that rely on the quality of the table contents. Unsurprisingly, the tables in WikiTables are just as small as those in WebTables.

Table 1: Existing large-scale relational table corpora and GitTables. Tables extracted from CSV files have considerably higher dimensions, evoking actual tables in databases.

| Name | Table source | # tables | Avg # rows | Avg # cols |
|------|--------------|----------|------------|------------|
| WebTables [23] | HTML pages | 90M | 11 | 4 |
| Dresden Web Tables [15] | HTML pages | 59M | 17 | 6 |
| WikiTables [4] | Wikipedia entries | 2M | 15 | 6 |
| Open Data Portal Watch[1] [28] | Open Data portals | 1M | 379 | 14 |
| VizNet [17] | WebTables, Plotly, i.a. | 31M | 17 | 3 |
| GitTables | GitHub repositories | 1.7M | 209 | 25 |

**Open Data Portal Watch** With 1M structured data files extracted from 260 Open Data portals [28], this is the first substantial corpus not sourced from HTML pages. Its large proportion of CSV files from mostly governmental institutions indicates that such files are frequently used in practice. From this corpus, 100K CSV files were parsed to tables and analysed for formatting, structure and data types [26]. This analysis illustrates the different dimensions and range type distributions of such tables, motivating the construction of a large-scale corpus like GitTables.

**VizNet** VizNet was constructed to train and evaluate visualization methods with real-world tables [17]. It combines 31M tables from WebTables [8], ManyEyes [38], Plotly [30], and Open Data Portal Watch [28]. A t-SNE projection of featurized tables suggests that tables not from the Web exhibit different internal structures from both WebTables and synthetically generated tables, providing further evidence for differences between Web tables and tables from other sources.

## 2.2 Table datasets with annotated column semantics

**T2Dv2** This small subset of tables from WebTables was built to benchmark methods for knowledge base (KB) augmentation [33]. The rows, columns and tables were manually annotated with correspondences to DBpedia instances, properties and classes. This was found to be a trivial target for KB matching due to the many "obviously" linkable entities. Recent work also points out that Web tables might not be typical of tables used for KB augmentation [11].

**SemTab** The SemTab challenge [19] provides datasets for benchmarking KB matching methods. Each dataset consists of synthetic tables and represents a different difficulty level for KB matching. The tables were populated by WikiData and "refined" by adding for example noise. SemTab 2020 incorporates 180 larger-sized Wikipedia tables were enriched with noise to mimic real tables [11]. Columns were annotated by linking cell entities to DBpedia types and aggregating these types to a column-level annotation. GitTables can be a useful resource in future rounds of the SemTab challenge for benchmarking table interpretation methods on database-like tables.

**TURL** Inspired by pretrained language models (e.g., [14, 32, 6]), TURL [13] provides a framework for learning embedding representations of Web tables through a pretrained model. Pretrained models require fine-tuning with labeled data to be applied to specific tasks in domains. To do this, a set of tables from WikiTables is annotated with 255 semantic types from Freebase. Application of learned table representations for table understanding motivates the construction of a larger-scale and rich corpus to support this nascent line of research.

Table 2: Characteristics of annotated relational table datasets. Existing annotated corpora are limited in the number of tables and types, while GitTables provides a large-scale corpus annotated with 2K types.

| Dataset | Table source | # tables | Avg # rows | Avg # cols | # types | Ontology |
|---------|--------------|----------|------------|------------|---------|----------|
| T2Dv2 [1] | WebTables | 779 | 17 | 4 | 275 | DBpedia |
| SemTab[2] [19] | WikiData, Wikipedia | 132K | 224 | 4 | - | DBpedia |
| TURL [13] | WikiTables | 407K | 18 | 3 | 255 | Freebase |
| GitTables | GitHub | 1.5M | 209 | 25 | 2K | DBpedia, Schema.org |

---

[1]Average # rows and columns are based on the analysis of 100K CSV files [26]
[2]We aggregate statistics of all table subsets of SemTab 2020.

# 3 GitTables

In this section, we first summarize the criteria that guided our curation of GitTables. We then discuss how GitTables satisfies these criteria and describe our corpus pipeline in detail.

## 3.1 Design principles of GitTables

Based on the gaps reported in the literature and our experience developing learned models for table interpretation tasks, we identified four criteria for GitTables:

**C1** To facilitate data-driven relational table models, the corpus should have a scale far exceeding the 1M structured data files living in Open Data portals.

**C2** To advance research beyond the Web, we need "database"-like tables which are considerably larger and more heterogeneous than the small, text-heavy Web tables.

**C3** The corpus should have topical coverage and content that generalizes to enterprises, governments, and medical institutions.

**C4** The increased interest in data-driven methods for e.g. data search and validation means that tables must be enriched with semantic annotations.

We considered different public interfaces to retrieve structured data files for our relational tables. On GitHub, we performed a simple search for CSV files containing the comma character (","), yielding 69,566,182 files[3]. This suggested that GitHub can be an effective resource for collecting a relational table corpus at scale (criterion **C1**). We focused on the CSV format given its wide use in storing raw structured data [26], as reflected by the massive number of CSV files we found in our search.

GitHub is commonly used by programmers, data scientists, and researchers, among others [29, 20]. Given the proximity of such users to actual databases, GitHub is a rich source for heterogeneous tables. Prior analyses of CSV files from GitHub also found that these files have diverse formatting and the tables extracted from them have relatively large dimensions [37, 20]. These properties are common across database contexts [37, 21], so that we consider CSV files from GitHub a suitable resource for database-like tables (**C2**).

Inspired by the construction of ImageNet [12], we selected 67K unique English nouns from WordNet [36] yielding a set of diverse keywords (called "topics") to specify our search queries. This introduces some bias towards English tables and skews the range type distribution (e.g. `Text` and `Number`) for some topics, since some topics such as "person" are more likely to appear in textual tables. However, we found that topics like "id" appear mainly in headers of numerical tables. In total, WordNet nouns have the desired topical coverage to ensure content diversity in GitTables (**C3**).

To satisfy the criterion **C4**, we provide semantic annotations for table columns. We developed two annotation methods, one a syntactic method informed by Sherlock [18] and the other leveraging a pretrained semantic model. We considered multiple ontologies to accommodate different use-cases, and selected DBpedia [2] and Schema.org [16] as they are well curated and provide complementary and diverse semantic types (as discussed further in Section 3.4).

Our high-level pipeline comes down to 1) extracting CSV files from GitHub, 2) parsing CSV files to tables, and 3) annotating tables with column semantics. Figure 2 visualizes this pipeline.

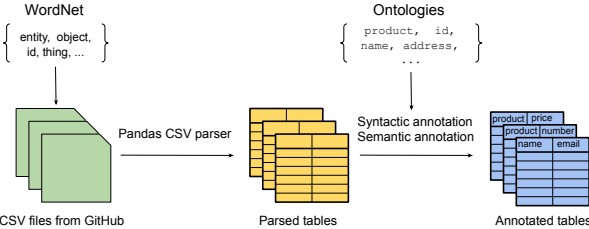

Figure 2: The pipeline for creating GitTables consists of 1) extracting CSV files from GitHub, 2) parsing CSV files to tables, and 3) annotating tables with column semantics.

---

[3]Retrieved 29 May 2021 from `https://github.com/search?p=100&q=%22%2C%22+extension%3Acsv&type=Code`

## 3.2 Extracting CSV files from GitHub

GitHub restricts querying in multiple ways to avoid overloading its Search API. First, it is not possible to retrieve files larger than 438 kB; this bounds the CSV files we extract. Although some organizations may use larger files, most CSV files in Open Data portals are found to be smaller than 100 kB [26]. Second, the search responses are limited to 1000 files. This restriction makes the process of extracting a massive set of CSV files nontrivial, as detailed below.

First we construct an initial "topic query" for each topic from WordNet restricted to files with the CSV format. For example, we retrieve CSV files that contain the word "object" by the query q=``object'' extension:csv. We execute this initial topic query through the GitHub Search API and get the initial response size of this query representing the number of GitHub URLs pointing to CSV files containing the word "object".

Since the API restricts the number of files per query to 1000 and many topic queries return around 100K files, we segment the initial queries. We use the "size" qualifier to perform this segmentation, and generate sequences of file size ranges (in bytes) proportional to the number of files in the initial response. This results in segmented topic queries like q=``id'' extension:csv size:50..100, q=``id'' extension:csv size:100..150, and so on. We execute all segmented queries and collect the paginated responses, each of which contains approximately 1000 URLs.

We then traverse the paginated responses to extract all URLs for a given topic. We iteratively write the raw contents pointed to by the URLs to CSV files, directly through HTTP requests.

## 3.3 Parsing CSV files to tables

Once we have the CSV files, we attempt to parse them to tables using the CSV parser from the Pandas library, a widely used data processing and analysis library for Python [25]. We leverage the integrated functionality of Python's Sniffer tool to determine the delimiter of the CSV files. We assume that all first rows correspond to the header row, as is typically the case for CSV files [35]. Lines at the beginning of the file are skipped in case they are empty. We also remove rows in case they are "bad lines," such as lines with an extra delimiter.

## 3.4 Annotating tables with column semantics

Models trained on existing datasets often make errors when it comes to non-mutually exclusive semantic types [18]. We might for instance find the same values in columns with the type annotations id and product id. Manual annotation aims at normalizing such cases, but manual annotation does not scale. We solve this problem by providing hierarchical relations for semantic types. For example, the type product id and tax id in Schema.org are all labeled as subproperties of identifier. This information can be exploited in model evaluation and training.

The opposite holds as well. We might encounter different type of values labeled as name, e.g. columns with company names and with last names. The surrounding context of columns is helpful for disambiguation [42, 13], but GitTables has enough columns per type to infer that types can be represented by different values. Also, if there is a semantic type for company name, the hierarchical relationship between the general and more fine-grained type can be used for disambiguation.

**Semantic types** In total, we extracted 2831 properties from DBpedia that we use as semantic types. From Schema.org we included properties as well as types that together sum to 2637 semantic types. Table 3 presents additional statistics on the ontologies. We provide the following metadata per semantic type if available:

1) semantic column type in English, e.g. id and name,
2) range type, e.g. Number and Text,
3) domain, e.g. address has domain Person, Organization, among others,
4) superclass or superproperty, e.g. product id → id, and
5) description, e.g. "The identifier property represents any kind of identifier for any kind of Thing, such as ISBNs, GTIN codes, UUIDs."

Figure 3 illustrates the difference in topical coverage between the two ontologies. Most semantic types from DBpedia relate to domains like Person, Place or PopulatedPlace while types in

Table 3: Counts of unique semantic types, range types, and domains in DBpedia and Schema.org.

| Ontology | # semantic types | # range types | # domains |
|---|---|---|---|
| DBpedia | 2831 | 189 | 274 |
| Schema.org | 2637 | 268 | 354 |

Schema.org are more scattered across domains and are topped by `CreativeWork`, `Organization`, `Person`, `Offer`, and `Product`. Another difference between these ontologies is the metadata provided along with semantic types. Schema.org for example provides descriptions for 100% of its semantic types.

**Annotation** Column headers are often formatted in different ways. For example, it was found that 50% of the column names contain underscores or were combined from multiple words into a single word using Camel Case formatting [26]. This informs our string preprocessing pipeline for column headers as well as semantic types. To summarize, we replace underscores and hyphens, split Camel-Cased combined words, and finally convert all strings to lower case. We discovered in experiments that column names with numbers were often annotated with semantic types from Schema.org that coincidentally contain a number. Given the scale of this behavior, the annotation pipeline does not annotate columns with numbers.

The original column names can be useful indicators of what a column's data consists of. To provide relatively strict annotation, we leverage the preprocessed column names and try to link them syntactically to semantic types in the ontologies, which is a common approach [18, 42]. Although column names are not normalized (the same column values might be named differently by different data creators), training and evaluation methods can incorporate the hierarchical information to account for this. We call this the syntactic annotation method.

Recent successes in semantic language models create the opportunity to annotate columns taking semantics into account. We use FastText [5] to embed column names and semantic types, and match them to each other. We use the character-level n-gram FastText model pretrained on the Common Crawl corpus. We experimented with contextual models like BERT [14] and the Universal Sentence Encoder [9], but found that these models yield worse or at best similar results. To obtain the final annotation, we take the match with the highest cosine similarity. Although users of GitTables can decide on a similarity threshold relevant to their tasks, we discard annotations with a similarity score lower than $\mu - 1\sigma$, where $\mu$ and $\sigma$ are the mean and the standard deviation of cosine similarities. The

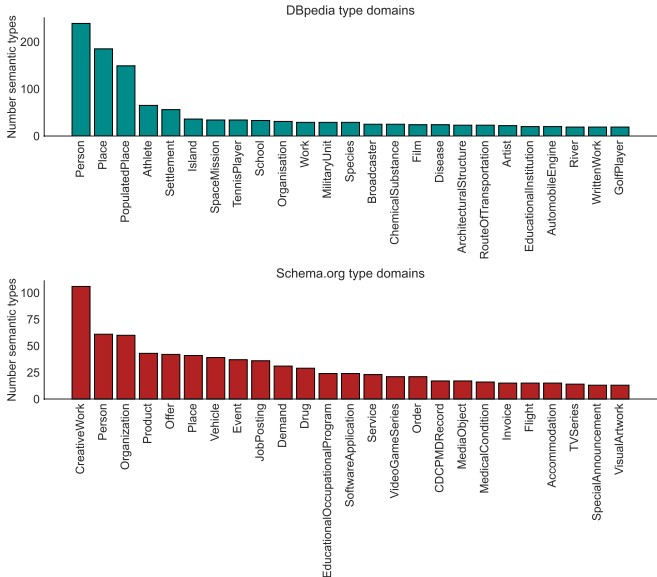

Figure 3: Top 25 domains of DBpedia and Schema.org in number of types contained.

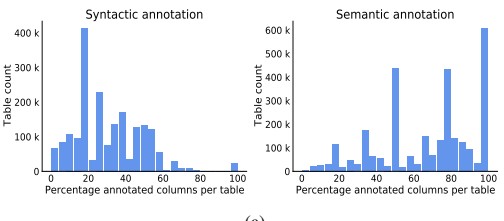
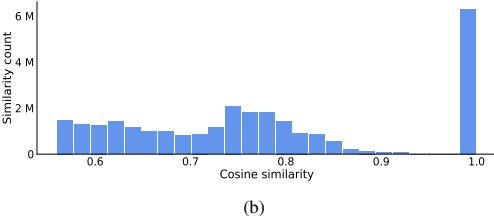

<p align="center">(a)                 (b)</p>

Figure 4: (a) Distributions of the percentage annotated columns per table. The semantic method yields on average more annotations per table. (b) Cosine similarity of annotated columns. The peak at 1 indicates syntactic similarity.

thresholding ensures that the annotations in GitTables are useful out of the box. We call this method the semantic annotation method.

## 4 Analysis

Once complete, our corpus will consist of 10M tables. To assess the topical differences and coverage implied by our topic-driven search queries, we analyse 10 topic subsets: "physical entity", "abstraction", "thing", "object", "whole", "living thing', "organism", "parent", "dwarf" and "id"[4].

### 4.1 Corpus statistics

The total analysis set of 1,717,883 tables consists of 358,834,439 rows and 42,722,012 columns, on average 209 rows and 25 columns. These dimensions differ significantly from Web table corpora with on average 5 columns and 15 rows. Also, the dimension of the tables itself are a better representation of database-like tables. These properties contribute to criteria C1 and C2. We observed the advantage of separating the tables from the query topics. For example, the query for "organism" tables yields many tables related to biological and medical entities, of which a typical one is shown in Figure 1. Such subsets can be leveraged for training domain specific models, or incorporate these topics as semantics for table embeddings.

The syntactic method annotates roughly 1M out of 1.7M tables (with at most 1 column annotated). This yields on average 33M annotations with 1K unique labels. The semantic annotation pipeline annotated on average 1.5M out of 1.7M tables, yields in total 139M column annotations with 2.6K unique labels. Statistics with regard to different ontologies are presented in Table 4. Depending on the use-case, users can select a suitable ontology and annotation method to select the relevant tables. If column context is of importance, for example for contextual models, having a high table-coverage is key. Figure 4a shows how many columns per table were annotated per method (aggregated over ontologies due to little difference). The similarity scores reflect the confidence of the semantic annotations. From the cosine-similarity distribution of the semantic annotations in Figure 4b, we conclude that many annotations have a similarity score around 1, which indicates syntactic similarity, while the remaining distribution is centered around 0.7. Users of the corpus can set the desired threshold based on their needs.

Table 4: Statistics of annotations by annotation method and ontology.

| Method | Ontology | # annotated tables | # annotated columns | # types | # types (n>1K) |
|---|---|---|---|---|---|
| Syntactic | DBpedia | 1,0M | 3,3M | 1218 | 309 |
| | Schema.org | 1,5M | 2,9M | 924 | 224 |
| Semantic | DBpedia | 1,0M | 13M | 2,7K | 1,2K |
| | Schema.org | 1,5M | 14M | 2,6K | 1,3K |

---

[4]Topics were chosen by order of the WordNet hierarchy. "id", which appears in WordNet as well, was added beyond this order as it is common in relational tables. We only extracted 10% of the tables for this topic to date.

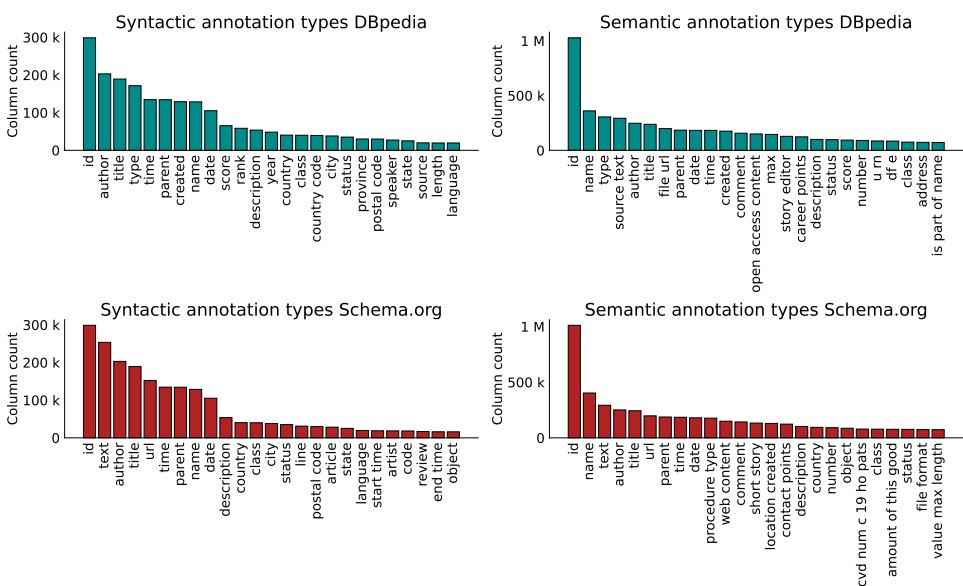

Figure 5: Frequencies of top 25 column semantic types in GitTables across two annotation methods, syntactic and semantic, and two target ontologies, DBpedia and Schema.org.

## 4.2 Table content and topical coverage

Beyond structural properties of tables in GitTables, we compare their contents to tables from VizNet which mainly stem from WebTables [17]. We interpreted the comparison of table contents as a data shift detection problem and evaluate if the data distributions significantly differ by training a domain classifier [31]. For this, we randomly sampled 10K columns from each corpus and extracted features as used for training Sherlock [18]. In total, we extract 1,188 features consisting of column-level statistics like column entropy and skewness, aggregations from word-embeddings, and aggregated character-level statistics (e.g. the number of hash (#) characters per cell).

We then train a Random Forest classifier with 100 estimators to separate whether a column originated from VizNet or from GitTables. Using a 10-Fold cross-validation setup, we find that this domain classifier is able to predict for 90% ($\pm 0.03$) of the columns from which corpus it originates. This separability of the corpora indicates the different data distributions in GitTables and VizNet, hence complementary value of GitTables.

The columns in the WebTables 2012 corpus have also been matched to DBpedia which resulted in the top 10 semantic types: `name`, `date`, `title`, `artist`, `description`, `size`, `type`, `location`, `model`, and `year`. For GitTables, the topical coverage and diversity of semantic types per annotation method and ontology is illustrated in Figure 5. Although top types in both corpora overlap such as `name` and `title`, we observe clear differences with top types in GitTables like `id`, `url`, `number` and `time`. Such types are more common in enterprise databases, and especially with `id` topping this list, the distribution of semantic types in GitTables shows that it meets criterion C3.

## 4.3 Annotation quality

As we have no access to the ground truth of semantic column types, we use the T2Dv2 benchmark hand-labeled with DBpedia types to evaluate our annotation pipeline [1]. Although the annotation quality on T2Dv2 does not ensure the quality of the annotations of GitTables itself, it is a good proxy for the quality of the annotations our pipeline produces. We considered table columns from files we could parse and that were annotated by T2Dv2 as well as the respective annotation pipeline. In total, we find 321 columns to evaluate for the semantic pipeline and 187 for the syntactic pipeline.

We find that our semantic pipeline yields in 54% (173 columns) the same annotation as T2Dv2. From the incorrect annotations, our pipeline annotated 47% (69 columns) with a DBpedia type that syntactically matches the column name as the corresponding similarity scores are 1.0. For example, our semantic pipeline annotated a column with cities (e.g. Pittsburg, Buffalo) named "City" with

the type `city` while T2Dv2 annotated this column as `location`. This motivated a manual review of the 148 columns for which we find different annotations between the semantic pipeline and the annotations from T2Dv2[5]. Based on our reviews (n=3), we find that on average in 63 ($\pm$ 14) out of 148 the semantic pipeline yields better annotations, in 37 ($\pm$ 3) out of 148 the T2Dv2 annotations were clearly better. In 33 columns ($\pm$ 14) they were just as good or bad, and undetermined in 15 cases ($\pm$ 17).

The syntactic pipeline yields in 61% (114 columns) the same annotation as T2Dv2. At first glance we found occurrences where the syntactic annotations were better. For example, a column with Latin names of birds named "Latin name" was annotated as `synonym` instead of `latin name` which is a type in DBpedia as well. We followed the same manual review process as for the semantic pipeline to review 73 annotations that were differently annotated. Based on inter-annotator agreement we find that the syntactic pipeline clearly yields more accurate annotations for 21 columns and T2Dv2 had better annotations for 9 columns.

These findings indicate that our annotation pipelines provide the necessary quality and flexibility for training, tuning, and evaluating DL models (C4). We also observe that the annotation quality of the T2Dv2 benchmark might require a review and potentially revision in future work.

### 4.4 Semantic column type detection

To illustrate the effectiveness of using GitTables we trained a semantic column type detection model on tables from GitTables. For this experiment, we selected five semantic types `address`, `class`, `status`, `name`, and `description` and randomly sampled 2K columns per type from our analysis subset. For each column we extracted the same features as before. We trained Sherlock using a 5-Fold cross-validation setup. This model achieves a macro F1 score of on average 0.82 ($\pm$ 0.01) illustrating its use for semantic column type detection.

We then trained a model on VizNet [17] using the same features, model and evaluation setup. To make a fair comparison, we sampled 2K columns of the exact same types being `address`, `class`, `status`, `name`, and `description`. This model yields a macro F1 score of 0.90 ($\pm$ 0.01) on samples from VizNet. However, evaluating this model on the columns of the same semantic types from GitTables, we find that it poorly generalizes as it only achieves a macro F1 score of 0.62 ($\pm$ 0.05). This illustrates the complementary data distribution of GitTables that is not well represented in Web table corpora.

## 5 Corpus usage

### 5.1 Content and structure

The parsed tables are stored in the Apache Parquet file format, which is widely used for efficiently storing and processing tabular data. Filenames are kept as found on GitHub but in case of filename duplication an identifier is added. The table metadata (e.g. atomic data types and URL of underlying CSV) and column annotations are stored in the metadata of the file. As each repository license imposes a unique set of rights to the tables that might be important for using the corpus, we added the repository license to the metadata of the files as well.

We publish the corpus in subsets of tables based on the topic used to query them, which enables the selection of table subsets depending on a user's needs. The released code[6] can be used for extracting tables from GitHub for custom query topics depending on the user's interest.

### 5.2 Accessibility

GitTables is hosted on Zenodo[7] which ensures long-term persistence of the corpus. We invite the community to enhance and curate GitTables further as well, and make requests to publish their new version if they do so. No issue will arise for users of the corpus who depend on a particular version

---

[5]The reviews can be found on `https://gittables.github.io`.
[6]`https://github.com/madelonhulsebos/gittables`
[7]`https://zenodo.org/record/4943312`

of GitTables, as Zenodo persists all versions. We will publish the larger corpus with approximately 10M tables as a new dataset on Zenodo.

Tables in GitTables were extracted from open-source GitHub repositories with licenses that allow distribution of the repository's content. We publish GitTables itself under the Creative Commons Attributions 4.0 International license (CC BY 4.0). All supplementary material such as code, documentation and the used ontologies can be accessed through the website of GitTables: `https://gittables.github.io`.

## 5.3 Responsible use

Despite the many positive applications that our work is intended to have, we recognize that publishing large uncurated datasets comes with potential risks. We identified two potential risks that we want to make users of GitTables aware of. First, it might occur that the CSV files underlying to GitTables, were posted unintended, contain sensitive or harmful information, or that the original files were removed for another reason. Making data files more accessible at scale may have a multiplicative negative impact if such undesired content is present. The spread and exact replication of undesirable content should be avoided, hence we call out to the community to report such observations so that we can remove these files accordingly.

Besides information spread it is known that training models on large uncurated datasets may impose biases [3]. As GitTables inherits the biases of its source, GitHub, it is predominantly English and corresponds to data from those who use Github. Models trained on GitTables might therefore be more accurate when applied to certain domains (e.g. business and science) or geographical regions where the web is more accessible. However, due to the nature of the data, relational tables, we do not foresee negative impact as a result from biased models towards certain subpopulations as identified for GPT-3 being trained on the Common Crawl corpus [3, 34]. But as such consequences can be unforeseen, it is important for users of GitTables to assess derived artefacts on the presence of such bias before deploying or publishing them. In case harmful biases are observed we would like to be notified so that we can mitigate these problems and improve our guidelines for using GitTables.

## 6 Conclusion

Relational databases are the bread and butter of consumer and enterprise applications. Prior corpora, dominated by Web tables, have successfully facilitated data-driven research into data management problems. While originally motivated by information extraction and retrieval tasks on the Web, these corpora have been used in unforeseen ways. Data management is at an inflection point to potentially benefit from recent developments in deep learning (DL), including deep transfer learning. The DL research and practice of the past decade illustrates that availability of high-quality data at scale is critical for the successful applications of DL. Similarly, having additional training data is also one of the most straightforward ways to improve the performance of DL on tasks in domains.

In this paper we introduce GitTables, extending the existing corpora of tables available to researchers. We extract the tables in our corpus from CSV files in open-source GitHub repositories and annotate their columns with semantic types from DBpedia and Schema.org. While this new corpus can be used in all the ways that prior tabular data corpora have been used, our primary motivation is to facilitate applications of learned models for relational data.

The use cases and analysis that we present here demonstrate only part of what can be done with GitTables and its extraction and annotation pipelines. Future work involves applying GitTables in additional use cases and getting feedback from the community to further enhance its usefulness. In the meantime, GitTables provides a realistic high-quality tabular dataset with annotations to help with efforts for improving and automating everyday data management tasks, including data analysis, relevant to consumer and enterprise systems.

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
