# OpenReview forum: "GitTables: A Large-Scale Corpus of Relational Tables"
_NeurIPS.cc/2021/Track/Datasets_and_Benchmarks/Round1 — Submitted to NeurIPS 2021 Datasets and Benchmarks Track (Round 1)_

### Official Review · Reviewer_4vtt · 2021-07-02
**This paper describes a large-scale scraped dataset of CSV files from Github for training systems that represent offline database tables.**

**Rating:** 7
**Confidence:** 3

**Strengths:**

Significance of contribution: Large. The paper makes a very convincing argument that this dataset is quite critical for extending high-capacity table-related models beyond simple HTML tables to offline relationship database tables. Specifically, the authors argue prior work has focused heavily on tables such as those found in Wikipedia that lack "ID" columns, large numbers of rows, and business-relevant content. GitTables represent an opportunity to extend the "table modeling" researching to many more domains.

Relevance to broader research community: Potentially high. The submitted manuscript describes in great detail a variety of specific applications, particularly at developing models that understand column semantics, are trained on relationship databases, etc. However, I imagine this dataset may be useful for a variety of other reasons, and there may be great interest in curating some subsets (e.g. for statistics, specific domain areas, etc.)

Ethical concern: The main paper does not discuss ethical (or legal) concerns in great depth but the supplemental materials do.

**Weaknesses:**

Relevance to broader research community: The submission as is may be difficult for the broader research community to use. Specifically, it seems a major requirement for this dataset to be used more widely will be the curation of smaller subsets. Many researchers may be concerned with training models or attempting to perform observational analyses on a massive dataset. However, this is certainly addressable and not necessarily a strong argument against publication in the paper's current form.

Ethics: a major weakness of the paper as submitted is a rather weak discussion of concerns around (1) licensing and (2) negative impacts of working with large, uncurated datasets (c.f. concerns raised in this high-profile paper: https://dl.acm.org/doi/10.1145/3442188.3445922). See below for more.

**Additional Feedback:**

Thanks to the authors for this submission!

**Clarity:**

The paper itself is very well written. Also of note is that the GitTables website is very cleanly presented, which will be relevant to users of the dataset.

**Correctness:**

The dataset preparation choices all seem reasonable. The paper includes fairly clear design principles which guide the choices. No major concerns with correctness.

**Documentation:**

Dataset creation is well documented. The choices related to column semantic annotations are explained well, and seem well motivated. I believe other researchers could replicate or extend this work based on the current manuscript.

Intended uses are clear and data is easily accessible.

The authors have prepared a separate website. Fittingly, this is hosted on GitHub itself. The dataset is released with a (CC BY 4.0) license. The data itself is hosted on Zenodo.

**Ethics:**

1. The current manuscript does not engage with the question of dataset licensing. While GitTables itself is licensed CC BY 4.0, there is no guarantee the constituent data has a license at all. I would strongly advocate for the main paper (not just supplementary materials) to more seriously warn potential dataset users about licensing concerns. My understanding is that this is a highly active area of discussion (which has been further amplified by the recent release of "Copilot" by GitHub itself) without current clear answers. Nonetheless, being more transparent about license concerns seems like a safe bet.
2. The current draft also does not mention any concerns around using a large, uncurated dataset. In the high profile "Stochastic Parrots" paper, the authors lay out concerns with "unfathomable" data, in particular relating to diversity, social views, and encoding bias. I believe that the data in GitTables could run into similar issues. Still, GitTables appears to me to make a major contribution to the research community. I think simply engaging with this concern in the main paper could help resolve this issue. Specifically, calling on users of GitTables to consider helping to curate the dataset and report on questions of data bias could be useful. For some applications of GitTables, this may not be an issue at all, but for others the concerns laid in Stochastic Parrots could be quite serious.

**Relation To Prior Work:**

Comparison with prior work is well written. It focuses specifically on "table corpora", but this seems quite reasonable relative to the contribution of this paper.

For future work, it may be interesting to compare GitTables with a large number of other dataset hosting services that also have many relational database-esque CSV files (e.g. Kaggle, Dataverse, etc.) but this is likely quite out of scope of the current submission.

**Summary And Contributions:**

[Edited Jul 19, 2021]:
Thanks to the authors for their response and for updating the submission. I think the revisions help to address the concerns raised in my review and the other reviews. After revisions, I continue to recommend acceptance.

-----

This paper describes a large-scale scraped dataset of CSV files from Github for training systems that represent offline database tables.

---

> ### Author Response · Authors · 2021-07-12
> **Response to your review**
>
> Dear reviewer,
>
> We thank you for your review, encouraging feedback and suggestions for improving our contributions. We address your feedback and suggestions in the sections “Responsible use”, “License compliance”, and “Accessibility and maintenance” in our common response.
>
> Please let us know if you think that any of your concerns or suggestions remain unaddressed.

---

### Official Review · Reviewer_Wm5b · 2021-07-03
**A large-scale dataset on github relational tables**

**Rating:** 6
**Confidence:** 4
**Correctness:** Mostly correct.
**Clarity:** Yes the paper is well-written.

**Strengths:**

- The proposed dataset, GitTables, is a relatively large-scale dataset (1.7M) up to date. The authors proposed an interesting approach of using distant supervision to annotate its semantic columns.

- The table is automatically extracted from Github, and based on the analysis this dataset shows a topical coverage and content structurally different from the web table corpora, thus could yield benefits in future applications.

- The authors presented detailed analysis on the topical distribution, syntactic and semantic types of column annotations, and demonstrated one use-case of semantic column type detection using GitTables.

**Weaknesses:**

- I think the authors should present a better analysis on how GitTables can be used. It is briefly discussed in Section 4.4, but it is not surprising that a model trained on web table corpora does not generalize very well on GitTables, since the topic distribution is different. I think it would be more convincing if the authors can show a use-case which is achievable by GitTable (given its wider coverage over certain topics) but not by web table corpora.

- A discussion on the responsible use of GitTables should be included. Given the skewed topic distribution of GitTables, will it yield any issues when training a model on this corpus? E.g., what kind of applications will GitTables be suitable for, and for what applications should we not use GitTables?

**Additional Feedback:**

Please add a discussion on the ethics/responsible use of GitTables. Is there any bias towards certain topics given the topic distribution on Github could be skewed?

**Documentation:**

There is sufficient detail on how the data is collected, processed, as well as annotated.

The data is available through the links provided by the author.

No maintenance plan is mentioned (the authors did mention a plan of growing it to 20M).

No ethical or responsible use is discussed (there's a very brief discussion in Section 3.1 on the bias towards English table, but more discussion/analysis is needed).

**Ethics:**

One thing that worth discussing is if there's any bias for a model trained on GitTables, e.g., Github might present a distribution skewed in certain topics, thus a model trained on this dataset might exhibit certain bias towards a few topics. More analysis/discussion on this front is needed.

**Relation To Prior Work:**

The relation to prior work is discussed adequately.

**Summary And Contributions:**

This paper proposes GitTables, a large-scale dataset of relational tables (1.7M), mined from GitHub. The corpus is annotated with > 2K semantic types using distant supervision.

The authors performed details analysis on GitTables, and found it has a topical coverage and content structurally different from web table corpora, thus potentially providing complementary benefits in future applications.

---

> ### Author Response · Authors · 2021-07-12
> **Response to your review**
>
> Dear reviewer,
>
> We thank you for your review and your valuable suggestions for improving our contributions. We would like to draw your attention to the sections “Responsible use”, “Accessibility and maintenance” and “Additional use-case” of our common response in which we address your feedback.
>
> Please let us know if you think that any of your concerns or suggestions remains unaddressed.

---

### Official Review · Reviewer_mYya · 2021-07-06
**An interesting and large corpus scraped from Github**

**Rating:** 6
**Confidence:** 2

**Strengths:**

* Gittables does solve some problems authors have highlighted in the related work section.
* Design principles described in section 3.1 are clearly identified and fulfilled by Gittables
* Authors have compared the distribution shift for this paper with the existing corpora and have presented that Gittables is a novel addition. Gittables can be helpful for training better models and will be helpful for the community.

**Weaknesses:**

* The authors have mentioned that they only scraped tables from public repositories, not open-sourced repositories. I think that Public repositories on Github are still subject to copyright law and according to GitHub's official policy. In the search query authors provided on the footnote of page 4, authors have queried all repositories instead of properly licensed repositories. Though authors have addressed this issue in supplemental materials.

* Authors have not mentioned the consequences of using the CSV files out of context. (this issue is addressed in the supplemental material though)

**Additional Feedback:**

No additional comments

**Clarity:**

The paper is very well written. Analysis and motivation behind the paper are very well described.

**Correctness:**

* There are no major issues with the correctness of the paper.

**Documentation:**

yes, there is sufficient detail to reproducibility.

**Ethics:**

* Licensing arrangement
* Using data out of context
These issues have been addressed in supplement data but not in the main paper.
In the supplement material, authors state that:
"All tables in GitTables were extracted from public GitHub repositories, hence it is assumed that publication of this data is not restricted", this might not be a fair assumption as the repositories without a license still come under copyright law.
"However, the original CSV files may be licensed and restricted in use, hence products built on GitTables inherit these licenses.  It is considered the responsibility of the user of the corpus to comply with these licenses and make responsible use of this data. "  This transfer of responsibility on the end-user is not very transparent and might not even be possible for end-user to comply with those licenses.

**Relation To Prior Work:**

The authors have clearly described how this work differs from prior work and have described novel contributions in the field from their work.

**Summary And Contributions:**

The authors in this paper present Gittables, which is a corpus of tabular data they have scraped from GitHub, annotated that large corpus of data analyzed, and presented this data for public use. The authors compared the dataset fairly with the present availability of corpora and have highlighted the importance of Gittables in a clear and concise manner.

---

> ### Author Response · Authors · 2021-07-12
> **Response to your review**
>
> Dear reviewer,
>
> We thank you for your review. We appreciate your recognition of the value of GitTables for the wider research community and your constructive feedback. We address your concerns in the sections “Responsible use” and “License compliance” in our common response.
>
> Please let us know if any of your concerns remain unaddressed.

---

### Author Response · Authors · 2021-07-12
**Response to our reviewers**

We thank our reviewers (R1=mYya, R2=Wm5b, and R3=4vtt) for their recognition of GitTables' value and their constructive feedback to improve the paper along with the corpus’ responsible use, compliance, and accessibility. Given the overlap of the feedback across our reviewers, we address the concerns raised in a common response below.

**Responsible use (R1, R2, R3)**

We agree with all reviewers on the importance of improving the visibility and clarity of our discussion on ethical risks, which was originally in the Societal Consequences section of the supplementary material. We move this section from the supplementary material to the main paper while revising it to bring attention to potential risks more concretely, including the risks of training high-capacity models on large uncurated corpora as discussed in [1]. Additional analysis of, for example, distributional properties of GitTables will inform this discussion. The additional page for the camera-ready version will allow us to expand our discussion further.

In addition to addressing these issues directly in the paper, we also revise our dataset website (https://gittables.github.io/) and the download page on Zenodo (https://zenodo.org/record/4943312)  to include notes on potential ethical risks.

**License compliance (R1, R3)**

We agree with R1 and R3 on the importance of taking into account the licenses of the repositories in our curation pipeline. To this end, we filter the corpus to contain only the tables from repositories with licenses that explicitly allow distribution. We adjust the target size of our corpus to be 10M to account for this filtering.

We also include the respective licenses in the table metadata to help users of the corpus select tables based on the correct licenses for their intended use. Our analysis of a subset of licensed tables indicates that the filtering will not significantly change the characteristics of our corpus.

**Accessibility and maintenance (R2, R3)**

Following the suggestion by R2 and R3, we revise our maintenance and accessibility description (originally in the supplementary material) and move it to the main paper. In our revision, we emphasize the corrective maintenance that we will carry out to ensure no sensitive information or explicit biases are present in GitTables. To this end, we also call out to the community through the dataset website and download page for reporting such observations.

Also, our revision further clarifies that users can select the (smaller) subsets of GitTables based on the intended scale and topics of interest. Similarly, we indicate that users can use our released code (https://github.com/madelonhulsebos/gittables) to extract tables based on custom topics of their interest beyond WordNet topics as well. This further facilitates flexible and diverse use of our work.

**Additional use-case (R2)**

As highlighted by R2, our paper illustrates how this corpus is different compared to Web table corpora. We would like to emphasize that the main value of GitTables, besides its annotations obtained using a scalable annotation pipeline, lies in the underlying data which better resembles database tables. Therefore, while GitTables supports use-cases similar to that of Web table corpora, it will help the models perform better on database tables.
In Section 4.4. we illustrate this point by evaluating a column type detection model trained on tables from the Web which poorly generalizes to GitTables although the semantic types are the same. We realize that the overlap of semantic types was not clearly stated; thank you for pointing this out. We clarified this in the revision.

We agree with R2 that it would be nice to demonstrate additional use-cases. We plan to evaluate the generalizability of models trained on GitTables as well as VizNet to a collection of hand-labeled database tables such as the Public BI Benchmark as used in [2]. We requested access to this labeled version of the Public BI Benchmark dataset. In this context, we’d like to reiterate the main motivation behind GitTables, the lack of publicly available, annotated corpora of database-like tables at scale.


[1] Bender, E. M., Gebru, T., McMillan-Major, A., & Shmitchell, S.  On the Dangers of Stochastic Parrots: Can Language Models Be Too Big? In Proc.  ACM Conference on Fairness, Accountability, and Transparency. 2021.

[2] Langenecker, S., Sturm, C., Schalles, C., & Binnig, C. Towards learned metadata extraction for data lakes. In Proc. BTW 2021.

---

### Comment · Reviewer_sw7e · 2021-08-06
**An official ethics review**

This is a comment by one of the official ethics reviewers.

The technical reviewers have rightly pointed out several concerns with the dataset. The license issue has been addressed by the authors already by conducting an additional filtering. That's great.

However, on the point of encoding bias, the authors state that "In our revision, we emphasize the corrective maintenance that we will carry out to ensure no sensitive information or explicit biases are present in GitTables." The future tense is the problem. Mitigating the release of sensitive information and mitigating unwanted biases is not so easy. I do not feel comfortable with the dataset being released and its paper published until the authors have taken some concrete actions and the results are seen.

---

> ### Author Response · Authors · 2021-08-10
> **Response to your ethics review**
>
> Thank you for bringing our attention to the difficulty of mitigating the spread of sensitive or harmful information and undesired bias after releasing the dataset. To address this concern, we follow existing literature and augment our pipeline, report results from new analyses, and update our revision accordingly.
>
> Here we first summarize our response under three themes and then provide details of our methodology and analysis findings.
>
> **Personal identifiable information**
>
> To minimize the risk of spreading sensitive personal data that was published on GitHub into GitTables, we follow best practices to handle personal data by anonymizing columns that might contain personal identifiable information (PII) [1].
>
> **Harmful content**
>
> Prior analysis of the Common Crawl corpus shows that content from social media platforms might contain harmful content [2]. To mitigate the propagation of such content, we remove tables that potentially contain harmful content from Twitter, Facebook, or Reddit.
>
> **Bias**
>
> Following the bias mitigation guidelines introduced by earlier work [3, 4], we analyze and report biases in GitTables so that users can take this into consideration in their downstream applications.
>
> *********
>
> In the following, we give details of our approach to implementing the above steps along with our findings, which we also include in our revision.
>
> **Methodology and Results (part 1)**
>
> In our final corpus, which contains tables from licensed repositories only, we mitigate the spread of sensitive information by anonymizing personal identifiable information (PII). We mapped a set of PII indicators, consistent with GDPR and NIST notions of PII [1], to semantic types from the DBpedia and Schema.org ontologies. Table 1 shows that, in total, approximately 0.35% of columns annotated with the Schema.org ontology and 0.50% of columns annotated with DBpedia ontology, correspond to one of the “personal identifiable semantic types” apart from the more general type “name”. Note that the superset type “name”  corresponds to roughly 2% of the columns, which does not necessarily indicate a person’s name. We mapped these personal identifiable semantic types to data classes from primarily the Faker library [5] (see Table 1) to replace values in the corresponding columns with fake values. We replace values in columns of tables where any of the personal identifier types “name” or “person” appears in combination with one of the other personal identifiable semantic types in Table 2. Given the relatively small number of PII columns, as shown in Table 1, this anonymization procedure does not significantly change the underlying data distribution of GitTables.
>
> Table 1: column percentages (based on 35K columns across 10 topics) per personal identifiable semantic type per ontology indicating the small proportion of PII data in GitTables, along with data classes to substitute values.
>
> | **Schema.org type** | **Corresponding columns** | **DBpedia type** | **Corresponding columns** | **Replacement class**     |
> |-----------------|-----------------------|--------------|-----------------------|------------------------|
> | name            | 2.202%                | name         | 2.041%                | faker.name             |
> | address         | 0.163%                | address      | 0.228%                | faker.address          |
> | person          | 0.068%                | person       | 0.065%                | faker.name             |
> | -               | -                     | age          | 0.0127%               | Random integer (0,100) |
> | email           | 0.042%                | -            | -                     | faker.email            |
> | birth date      | 0.017%                | birth date   | 0.014%                | faker.date             |
> | birth place     | 0.003%                | birth place  | 0.003%                | faker.city             |
> | postal code     | 0.003%                | postal code  | 0.020%                | faker.postal_code      |
> | home location   | 0.008%                | -            | -                     | faker.city             |
> | telephone       | 0.003%                | -            | -                     | faker.phone_number     |

---

> > ### Author Response · Authors · 2021-08-10
> > **Response to your ethics review (part 2)**
> >
> > **Methodology and Results (part 2)**
> >
> > Other sources of undesired content spread are user generated text (e.g., messages) on platforms on which biased opinions and hateful content might be openly shared. Analyses of raw web corpora like Common Crawl, illustrate that unstructured collections of texts from web pages include harmful content mainly originating from social media platforms [2]. As tables from GitTables stem from a platform for software developers and given their frequent use of APIs of social platforms like Twitter, this data appears in a small subset of tables. To ensure that no hateful content from these platforms is present in GitTables, we remove tables that have columns containing “tweet”, “twitter”,  “facebook”, or “reddit”, which account for 0.18% of the tables in total. We believe that this minimizes the risk of spreading harmful content without changing the characteristics of the corpus.
> >
> > We anticipate that GitTables will mainly be used for applications like data integration, augmentation, and search, similar to WebTables [6]. The analysis of the topical distribution present in GitTables shows a few semantic types that could potentially make methods informed by GitTables lean towards certain subpopulations, industries, or geographic areas. To further profile our dataset to this end, we adopted 3 categories as proposed in [7], being person, object, and geography. The topical distribution already presents the skew towards certain objects (e.g. with semantic types as id, type, time, date, description, rating, etc.). In Table 2, we present distributions along the geography and person dimensions informed by column values associated with relevant semantic types from DBpedia and Schema.org, which we report in our revision as a figure. This analysis confirms that tables in GitTables primarily represent English-speaking countries and cities. With regard to persons, the small number of columns in GitTables with semantic types that indicate population segments, e.g. gender, race, ethnicity, and nationality, indicate a relatively higher concentration of data representing Western countries. Our revision reports the findings to help users of GitTables take this information into account in their downstream applications.
> >
> > | **Category**   | **Semantic type** | **Percentage columns** | **Frequent values (most frequent first)**           |
> > |------------|---------------|--------------------|--------------------------------------------------------------------------------------------|
> > | Geography  | country       | 0.086%             | United States (merged with USA), Canada, Belgium, Germany          |
> > | Geography  | city          | 0.056%             | New York, London, Coquitlam, Cambridge                                                     |
> > | Population | gender        | 0.040%             | Male, Female, F, M                                                                         |
> > | Population | nationality   | 0.003%             | French, Dutch, Spanish, Mexican                                                            |
> > | Population | race*         | 0.007%             | Men, Human, White                                                                          |
> > | Population | ethnicity*    | 0.003%             | Hispanic, White, Caucasian (White), White - English/Welsh/Scottish/ Northern Irish/British |
> >
> > *Not present in Schema.org annotations.
> >
> > ******
> >
> > [1] Singhofer, F., Garifullina, A., Kern, M., & Scherp, A. (2021). rx-anon--A Novel Approach on the De-Identification of Heterogeneous Data based on a Modified Mondrian Algorithm. arXiv preprint arXiv:2105.08842.
> >
> > [2] Bender, E. M., Gebru, T., McMillan-Major, A., & Shmitchell, S. (2021). On the Dangers of Stochastic Parrots: Can Language Models Be Too Big? In Proceedings of the 2021 ACM Conference on Fairness, Accountability, and Transparency (pp. 610-623).
> >
> > [3] Mehrabi, N., Morstatter, F., Saxena, N., Lerman, K., & Galstyan, A. (2021). A survey on bias and fairness in machine learning. ACM Computing Surveys (CSUR), 54(6), 1-35.
> >
> > [4] Gebru, T., Morgenstern, J., Vecchione, B., Vaughan, J. W., Wallach, H., Daumé III, H., & Crawford, K. (2018). Datasheets for datasets. arXiv preprint arXiv:1803.09010.
> >
> > [5] Faker (2014), https://faker.rtfd.org. Online: accessed on 7 August 2021.
> >
> > [6] Cafarella, M., Halevy, A., Lee, H., Madhavan, J., Yu, C., Wang, D. Z., & Wu, E. (2018). Ten years of WebTables. Proceedings of the VLDB Endowment, 11(12), 2140-2149.
> >
> > [7] Wang, A., Narayanan, A., & Russakovsky, O. (2020). REVISE: A tool for measuring and mitigating bias in visual datasets. In European Conference on Computer Vision (pp. 733-751). Springer, Cham.

---

> > > ### Comment · Reviewer_sw7e · 2021-08-10
> > > **thank you for your efforts**
> > >
> > > Thank you for your great effort in doing all of these items in such a timely manner. I am now satisfied.
> > >
> > > In addition to reporting all of the stuff you did and the statistics in the paper, you may consider creating and posting an official factsheet that is prominently displayed on the website.

---

> > > > ### Author Response · Authors · 2021-08-10
> > > > **Thank you for your feedback and suggestion**
> > > >
> > > > Thank you for your prompt feedback and suggestion to compile a factsheet for GitTables.
> > > >
> > > > We will add a factsheet to our website as suggested to convey the data characteristics and use-cases to users of GitTables.

---

### Comment · Reviewer_gX7Z · 2021-08-12
**An Official Ethics Review**

The ethical concerns mentioned throughout the reviews and comments, which concern PII, licenses, responsible use, skews, and harmful content, have been reasonably well-addressed given the state-of-the-art in automatically handling these issues within datasets.

I agree with the previous ethics reviewer that a factsheet/datasheet/data card documentation would be a boon; this is also where the authors could further share the concerns raised and how they have addressed them (which could be a model for other dataset releases, IMO).

The largest remaining issue I see is that this dataset is shared as fully extracted content, rather than as urls.  When data is constructed as a set of urls, connected to the original source, then it is much easier for people who don't want their data in the dataset to simply delete their original sources and have that percolate to any further use of the dataset.  Ideally, the part of this dataset that is the table itself with columns, rows, cell values and a header with the original column names, could instead be a *link* to this content, where the content behind each link undergoes regular auto-population updates from the original sources that still return valid content.  This is a lot more work that opens up a lot of further issues (such as whether earlier versions of the dataset used to train deployed models should still be "around" in some way), so is a longer-term goal.

Although this dataset includes ethically problematic content, including the use of extracted (rather than linked) content, and imperfectly identified PII, etc., the authors have addressed the issues as best they could given current state of the art and the dataset's intended and foreseeable uses.  The dataset appears to be releasable if the work is accepted, but I encourage the authors to consider a url-only release in the future.

---

> ### Author Response · Authors · 2021-08-13
> **Thank you for your review**
>
> Thank you for your feedback and suggestion to consider a url-only release in the future.
>
> To make it easy for data owners to have their data removed from the corpus or anyone to inform us about undesired content, we provide a request form embedded in our website. We revised the text accompanying the form to make this intended usage of the form more explicit.
>
> For future releases, we will review alternative approaches, including url-only ones, to further facilitate the effectiveness of data removal, while ensuring accessibility along with our ability to anonymize PII and suppress hateful content.

---

### Comment · Reviewer_r9J4 · 2021-08-13
**Additional comments after reading the prior discussion**

As mentioned by the previous reviewers, I believe that a data statement/datasheets/factsheet etc would be good to accompany this work. In addition, I have the following comments below.

“Inspired by the construction of ImageNet [12], we selected 67K unique English nouns from WordNet [36] yielding a set of diverse keywords (called “topics”) to specify our search queries”

Papers like [1] show some of the issues with this approach, including derogatory images and labels in ImageNet and Tiny Imagenet. This resulted in the Tiny Imagenet dataset being taken down. What was done here to ensure that such categories are not included? I see that you have filtered out twitter/facebook etc below. Also note that automatic filtering out of certain words can in itself be harmful to members of marginalized groups [2].

“In total, we extracted 2831 properties from DBpedia that we use as semantic types.”
How did the authors decide to use these specific properties? What are some potential issues with these properties (e.g. see above regarding works pertaining to WordNet) and how were they mitigated? What are the potential issues of annotating columns using models such as FastText pertained on common crawl?

Have the authors done an analysis of the topic subsets besides the 10 mentioned in the paper? Given the scale of the dataset I am concerned about the same issues as mentioned in [1]. As mentioned in [2], curating datasets and documenting them is key. Have the authors attempted to create data statements (or data sheets) for this dataset? My understanding was that it was required for this NeurIPS track but I might have misunderstood.

[1] Birhane, Abeba, and Vinay Uday Prabhu. "Large image datasets: A pyrrhic win for computer vision?." 2021 IEEE Winter Conference on Applications of Computer Vision (WACV). IEEE, 2021.

[2] Bender, Emily M., et al. "On the Dangers of Stochastic Parrots: Can Language Models Be Too Big?🦜." Proceedings of the 2021 ACM Conference on Fairness, Accountability, and Transparency. 2021.

---

> ### Author Response · Authors · 2021-08-15
> **Thank you for your review**
>
> Thank you for your review and for highlighting potential effects of certain taxonomies (e.g. WordNet) for the expanded version of GitTables.
>
> **WordNet topics**
>
> We filter out topics from WordNet like “killing” and “naked” as tables associated with such topics are beyond the scope of GitTables. Our aim is not to build a table corpus with exhaustive semantic coverage but one representing databases typically found in enterprises, scientific institutions, governmental organizations, etc. We add this refinement to the revision of the manuscript.
>
> **Annotation**
>
> DBpedia and Schema.org were chosen for their level of curation and the rich semantics they capture. The curation of these ontologies also result in exclusion of offensive properties and classes which we confirmed by manual review of the semantic types. The semantic types enabled the analysis of biases and anonymization of PII data present in the tables, which we include in our revision.
>
> **Documentation**
>
> In the manuscript we analyze all tables, corresponding to the 10 mentioned topics, present in the current version of GitTables. We release this version as a separate version along with its factsheet. The future, expanded, version of GitTables will be extracted, curated and analyzed following the same procedures. We will compile a separate factsheet for that version to give full visibility in its topical coverage and potential biases.

---

### Note · ~Madelon_Hulsebos1 · 2021-06-08

https://zenodo.org/record/4943312#.YMcUlzYzZ4I

---

### Decision · Program_Chairs · 2021-07-26

**Decision:**

Reject

**Comment:**

Due to concerns raised by the ethics reviewers, we are not able to accept your submission at this stage. However, we warmly welcome you to update your manuscript according to the ethical comments and recommendations and resubmit to Round 2.